# Hierarchical uncertainty estimation for learning-based registration in neuroimaging

**Xiaoling Hu**[1,†], **Karthik Gopinath**[1], **Peirong Liu**[1], **Malte Hoffmann**[1], **Koen Van Leemput**[1,2],
**Oula Puonti**[1,3,‡], **Juan Eugenio Iglesias**[1,4,5,‡]

[1]Massachusetts General Hospital and Harvard Medical School
[2]Aalto University
[3]Danish Research Centre for Magnetic Resonance, Copenhagen University Hospital
[4]Hawkes Institute, University College London
[5]Computer Science and AI Laboratory, Massachusetts Institute of Technology

## Abstract

Over recent years, deep learning based image registration has achieved impressive accuracy in many domains, including medical imaging and, specifically, human neuroimaging with magnetic resonance imaging (MRI). However, the uncertainty estimation associated with these methods has been largely limited to the application of generic techniques (e.g., Monte Carlo dropout) that do not exploit the peculiarities of the problem domain, particularly spatial modeling. Here, we propose a principled way to propagate uncertainties (epistemic or aleatoric) estimated at the level of spatial location by these methods, to the level of global transformation models, and further to downstream tasks. Specifically, we justify the choice of a Gaussian distribution for the local uncertainty modeling, and then propose a framework where uncertainties spread across hierarchical levels, depending on the choice of transformation model. Experiments on publicly available data sets show that Monte Carlo dropout correlates very poorly with the reference registration error, whereas our uncertainty estimates correlate much better. Crucially, the results also show that uncertainty-aware fitting of transformations improves the registration accuracy of brain MRI scans. Finally, we illustrate how sampling from the posterior distribution of the transformations can be used to propagate uncertainties to downstream neuroimaging tasks. Code is available at: `https://github.com/HuXiaoling/Regre4Regis`.

## 1 Introduction

Aligning two or more images to a common coordinate frame, referred to as image registration, is one of the fundamental tasks in medical image analysis, especially in human neuroimaging with MRI. Since many human brain structures are fairly consistent across subjects, registration methods have been very successful in this domain. Registration plays a vital role in many important applications. One example is measuring temporal change in longitudinal studies by registering scans of the same subject at different time points (Holland et al., 2011): by using each subject as its own control, confounding effects introduced by inter-subject morphological variability are considerably reduced. Another important application has been image segmentation: prior to deep learning, the state of the art was established by multi-atlas approaches (Iglesias & Sabuncu, 2015), based on registering a set of labeled cases to the target scan and merging the deformed segmentations. Yet another key application has been spatial normalization to a digital atlas (e.g., the ubiquitous MNI template, Fonov et al. 2009), which is at the core of many neuroimaging studies, and which enables analyses (e.g., regression, group comparison) as a precise function of spatial location (Sowell et al., 2003).

Classically, image registration is cast as an optimization task, where the aim is to maximize a measure of similarity between a pair of images with respect to a (non)-linear transformation – often combined with a regularization term that prevents excessively convoluted deformations (Zitova & Flusser, 2003). This problem is typically solved with standard numerical optimization methods

---

† Email: Xiaoling Hu (xihu3@mgh.harvard.edu); ‡ Co-senior authors

(Nocedal & Wright, 1999). In medical imaging, the different components of optimization-based registration have been exhaustively studied, including transformation models (Rueckert et al., 1999; Christensen & Johnson, 2001), similarity metrics (Pluim et al., 2003; Avants et al., 2008), and optimization approaches (Klein et al., 2007; Glocker et al., 2011). The reader is referred to (Sotiras et al., 2013) for a comprehensive survey.

During the last decade, the focus on image registration has shifted from optimization-based approaches to (deep) learning-based approaches. By sidestepping the numerical optimization task, these algorithms can predict a mapping between two images almost instantaneously. Earlier deep learning (DL) methods, best represented by QuickSilver (Yang et al., 2017), were trained in a supervised manner using ground-truth deformation fields obtained with classical optimization-based registration. These supervised approaches were superseded by unsupervised frameworks that learn to directly minimize the dissimilarity between two input images, using losses similar to those of classical methods – possibly combined with segmentation losses for improved alignment of anatomical regions (Balakrishnan et al., 2019; De Vos et al., 2019). By now, many features of classical registration, including diffeomorphisms (Krebs et al., 2019), symmetry (Iglesias, 2023), progressive warping (Lv et al., 2022), or inter-modality support (Hoffmann et al., 2021) have been incorporated into DL registration.

A crucial, but less explored, aspect of image registration is uncertainty estimation. Measures of uncertainty can enhance both the reliability and interpretability of the registration process, which are crucial aspects in many downstream applications. In high-stakes clinical applications, such as surgery or radiation therapy, uncertainty estimates can be used to highlight regions with potential registration errors, e.g., when registering pre- and intra-operative images (Simpson et al., 2011a), or to give more reliable estimates of structure borders, e.g., when estimating margins around organs at risk in radiation therapy (Risholm et al., 2011). In neuroimaging studies, registration errors propagate to downstream analysis tasks, such as segmentation, group-level statistics, or functional mapping, and uncertainty estimates can be used to weigh or exclude data from unreliable regions to compute e.g., improved spatial statistics (Simpson et al., 2011b). In the classical registration literature, uncertainty models often rely on formulating the task as a probabilistic model, where the transform is a random variable (Simpson et al., 2013; Le Folgoc et al., 2017; Risholm et al., 2013; Le Folgoc et al., 2016; Kybic, 2009; Agn & Van Leemput, 2019), and where the uncertainty is described by the posterior probability distribution of the transform. However, uncertainty estimation remains largely unexplored in the modern DL registration literature. While there are well-established approaches in the DL literature (particularly ensembling and Monte Carlo dropout, Gal & Ghahramani 2015; Lakshminarayanan et al. 2017), they operate at the voxel level – rather than on the whole transform. For this reason, uncertainty estimates in registration are scarce and typically a by-product of model predictions (Dalca et al., 2019b; Gong et al., 2022; Krebs et al., 2018).

Here, we propose to integrate uncertainty estimation into DL registration in a principled manner. Specifically, we propose a method for uncertainty-aware fitting of transformation models to predictions made *independently* at different locations (typically at each voxel), which can directly capitalize on existing DL uncertainty estimation approaches. In this framework, DL solves a simpler location-by-location regression task, where a network is trained to predict a deformation vector per location (or, alternatively, a triplet of target coordinates Gopinath et al. 2024), along with uncertainty estimates (aleatoric and/or epistemic). We can then fit multiple transformation models to the set of predictions; our methods are general and support, e.g., affine transforms, B-splines (Rueckert et al., 1999), or non-parametric (Avants et al., 2008) transforms. The uncertainty estimates can then be propagated to the model parameters in closed form, enabling: *(i)* a weighted fit, where uncertain locations contribute less to the fitting; and *(ii)* uncertainty estimation of the model parameters, e.g., B-spline coefficients. The uncertainty on the model parameters effectively considers dependencies across spatial locations and can be further propagated to downstream tasks, e.g., registration-based segmentation.

In sum, our method models uncertainty as it propagates through a hierarchy of levels (network output, transform models parameters, downstream tasks), in a principled way that enables sampling, investigating modes of variation, and computing of error bars. Specifically, the main contributions of this work are:

1. We propose a framework for propagating network estimated uncertainties (epistemic and/or aleatoric) to transformation models and further to downstream tasks. Furthermore, our

     framework allows fitting of multiple different transformation models, which can be flexibly chosen based on the application without retraining the network.

   2. Using an experimental setup based on a recently proposed coordinate-regression DL method for atlas registration, we show that *aleatoric* uncertainty estimates correlate well with registration error but the epistemic uncertainty (estimated with Monte Carlo dropout) does not. Crucially, we also show that incorporating the aleatoric uncertainty into the fitting of the transformation model *increases* registration accuracy.

## 2 RELATED WORK

**Deep learning based medical image registration.** During the last decade, deep learning methods have dominated medical image registration (Cao et al., 2017; Krebs et al., 2017; Rohé et al., 2017; Uzunova et al., 2017; Sokooti et al., 2017; Tian et al., 2022; Balakrishnan et al., 2019; Dalca et al., 2019b;a;b). These methods directly predict a deformation field given two images, and can be generally categorized as supervised and unsupervised. Supervised methods train deep neural networks with ground truth deformation fields. For example, the pioneering QuickSilver method (Yang et al., 2017) uses fields estimated with an accurate, computationally expensive Large Deformation Diffeomorphic Metric Mapping (LDDMM) model. Cao et al. (2017) learns the complex mapping from the input patch pairs to their respective deformation field in a patch-based manner. In the context of prostate imaging, Krebs et al. (2017) investigates how deep learning could help organ-specific deformable registration, in applications such as motion compensation or atlas-based segmentation. Uzunova et al. (2017) seeks to learn highly expressive appearance models from a limited number of training samples.

Requiring ground truth deformation fields has the disadvantage that trying to learn the distribution of such fields in isointense image regions may lead to wasted model capacity and misguiding gradients due to overfitting. In contrast, unsupervised registration usually uses spatial transformer networks (STN) (Jaderberg et al., 2015) to warp moving images to match fixed images, and the model parameters are trained using the similarity between warped and fixed images. Similarly to classical methods, regularization terms are often used to encourage the smoothness of the predicted displacement fields. Representative unsupervised methods include VoxelMorph (Balakrishnan et al., 2019), its variational extensions (Dalca et al., 2019b), and (De Vos et al., 2019). These unsupervised methods achieve accuracy levels comparable with classical techniques, albeit with much higher efficiency.

**Uncertainty estimation.** Uncertainty estimation seeks to assess how confident a model is in its predictions – which is of great importance in the deployment of models in the real world, particularly in critical applications such as medical imaging. DL models deal with two types of uncertainty: aleatoric and epistemic. The former is input-dependent, e.g., noise in the data, and can be learned during training (Malinin & Gales, 2018). Epistemic uncertainty is on the model weights, e.g., due to insufficient training data. Principled formalisms such as Bayesian neural networks are possible, but are only practical for smaller models (Kendall & Gal, 2017). Recently, Monte Carlo (MC) dropout (Gal & Ghahramani, 2015) and model ensembles (Lakshminarayanan et al., 2017; Rupprecht et al., 2017) have been proposed as more practical approaches, especially in the context of classification tasks (Abdar et al., 2021; Gawlikowski et al., 2021). Unfortunately, these approaches also have shortcomings: MC dropout, which randomly turns off a fraction of the neurons, is fast and easy to implement but is known to underestimate uncertainty (Blei et al., 2017), whereas ensembles that estimate uncertainty as variance across multiple networks, or multiple output layers, are more accurate but computationally expensive in training (Lakshminarayanan et al., 2017).

**Uncertainty estimation for image registration.** Uncertainty estimation for medical image registration provides a layer of reliability and interpretability, and has long been a research objective. In the classical literature, methods based on probabilistic modeling have enabled uncertainty estimation via Bayesian inference (Simpson et al., 2013; Kybic, 2009; Le Folgoc et al., 2017; Risholm et al., 2013; Le Folgoc et al., 2016; Agn & Van Leemput, 2019). This is achieved by computing (exactly or approximately) the posterior probability distributions of the deformation model parameters. Instead of Bayesian inference, other methods have used bootstrap sampling as an empirical ensemble method to estimate registration uncertainty (Kybic, 2009).

In the DL era, registration uncertainty is often underutilized, and most existing approaches either rely on direct application of the general uncertainty estimation techniques described above (see for

instance Gong et al. 2022; Smolders et al. 2022; Chen et al. 2024a and Chen et al. 2024b for a survey), or obtain simplistic uncertainty estimates as a by-product (e.g., the variational inference strategy in Dalca et al. 2019b; Sedghi et al. 2019, which is known to underestimate uncertainty). Therefore, these approaches fail to consider the spatial distribution of deformation fields. Zhang et al. 2024 propose another version of an aleatoric loss for registration, which is parallel to the first level of uncertainty, see Section 3.1, in our framework. We propose to propagate the uncertainties further on to the transformation models and downstream tasks. Finally, we note that the correlation between uncertainty estimates and registration errors has not been thoroughly investigated (Luo et al., 2019; 2020) – possibly due to the scarcity of datasets with labeled pairs of landmarks that would ideally be used to measure these errors (Luo et al., 2020).

# 3 METHODS

Since our framework is applicable to different registration tasks, we first present it in general terms. Specific instantiations of the model, losses, and training approach are presented in Section 3.4.

**Preliminaries.** Given an input image, or input images, $\mathbf{I}$, our goal is to train a network that predicts $\mathbf{d} = [\mathbf{d}_1, \mathbf{d}_2, \mathbf{d}_3]$ where $\mathbf{d}_j = [d_{j,1}, ..., d_{j,N}]^T$ is a column vector storing $N$ values for the coordinate direction $j$. Depending on the application, the values may correspond to target $(x, y, z)$ coordinates, displacements from reference voxels in a target image that are needed to reach voxels in an input image, or even key points that can be used for landmark-based registration (Wang et al., 2023). We are thus seeking to train a neural network $\mathbf{F}_\theta$ with parameters $\theta$, such that: $\mathbf{d} = \mathbf{F}_\theta(\mathbf{I})$.

## 3.1 FIRST LEVEL OF UNCERTAINTY: TARGET REGRESSION

The strength of our method lies in its ability to capitalize on well-established uncertainty estimation methods that operate at the level of the individual outputs $d_{j,n}$. In this context, one can use the techniques discussed in Section 1 and Section 2 above to obtain an accurate estimate of the distribution of the field at each point individually. Without loss of generality, we assume this distribution to be Gaussian: Monte Carlo dropout and ensembles both yield samples that are typically summarized into a mean and (co-)variance; whereas aleatoric uncertainty estimation for continuous variables often relies on prediction of Gaussian means and (co-)variances as well (Tanno et al., 2017).

Therefore, and irrespective of the chosen uncertainty modeling approach, we assume throughout the rest of this manuscript the availability of $\boldsymbol{\mu} = [\boldsymbol{\mu}_1, \boldsymbol{\mu}_2, \boldsymbol{\mu}_3]$ and $\boldsymbol{\sigma} = [\boldsymbol{\sigma}_1, \boldsymbol{\sigma}_2, \boldsymbol{\sigma}_3]$, where $\boldsymbol{\mu}$ stores the predicted mean values for every position and direction and $\boldsymbol{\sigma}$ denotes the corresponding predicted standard deviations. Henceforth, we refer to these as the *first level of uncertainty*.

We further note that, since uncertainty is considered, overfitting in flat image regions is not an issue and one can safely train the network with supervised losses – which greatly facilitates learning of aleatoric uncertainty. In this case, ground truth coordinates or displacements can be obtained by registering images to other images (pairwise registration) or MNI (atlas registration) using a slow, accurate, classical method (Yang et al., 2017).

## 3.2 SECOND LEVEL OF UNCERTAINTY: WEIGHTED FITTING

Given a set of predicted values $\boldsymbol{\mu}$ and their standard deviations $\boldsymbol{\sigma}$, we can fit a large family of (non-)linear transformations, including models based on basis functions (e.g., B-splines, Rueckert et al. 1999) and non-parametric approaches (Thirion, 1998).

**Uncertainty-Aware Parametric Transformations.** Let $\boldsymbol{\phi} = [\boldsymbol{\phi}_1, ..., \boldsymbol{\phi}_B]$ be a matrix of $B$ basis functions where $\boldsymbol{\phi}_b = [\phi_{b,1}, ..., \phi_{b,N}]^T$ and $\phi_{b,n}$ represents the value of basis function $b$ evaluated at location $n$. Finally, let $\mathbf{c}$ be a $B \times 1$ vector with the coefficients of the basis functions. To fit the model, we minimize the coordinate error weighted by the corresponding precisions, given by the inverse variances. This can be solved in one coordinate direction $j$ at the time using standard weighted least squares. Given a weight matrix $\mathbf{W}_j = \mathrm{diag}(\boldsymbol{\sigma}_j^{-2})$, the goal is to minimize the weighted squared error $E_j$ for each of the three dimensions $j = 1, 2, 3$:

$$E_j = [\boldsymbol{\mu}_j - \boldsymbol{\phi}\boldsymbol{c}_j]^T \boldsymbol{W}_j [\boldsymbol{\mu}_j - \boldsymbol{\phi}\boldsymbol{c}_j],$$

which has the well-known solution:

$$\boldsymbol{c}_j^\mu = [\boldsymbol{\phi}^T \boldsymbol{W}_j \boldsymbol{\phi}]^{-1} \boldsymbol{\phi}^T \boldsymbol{W}_j \boldsymbol{\mu}_j = A \boldsymbol{\mu}_j,$$

where $\boldsymbol{c}_j^\mu$ is the mean of the fitted coefficients for coordinate direction $j$, and $A_j = [\boldsymbol{\phi}^T \boldsymbol{W}_j \boldsymbol{\phi}]^{-1} \boldsymbol{\phi}^T \boldsymbol{W}_j$ is the weighted pseudoinverse. Further, since this is a linear estimate, we can compute the $B \times B$ covariance matrix of the fitted coefficients:

$$\boldsymbol{c}_j^\Sigma = A_j * \boldsymbol{W}_j^{-1} * A_j^T. \tag{1}$$

We note that the basis function formulation covers both linear and non-linear transformations, as the former can be seen as a special case of the latter with $\boldsymbol{\phi} = [\mathbf{x}, \mathbf{y}, \mathbf{z}, \mathbf{1}]$, where the first three columns are the coordinates at the corresponding $N$ locations in the input image space and the last one is a column of ones. These linear basis functions can be fitted in isolation (linear registration), prior to nonlinear fitting (sequential linear / nonlinear registration), or together with nonlinear basis functions in a single $\boldsymbol{\phi}$ (joint linear/nonlinear registration).

Given $\boldsymbol{c}^\mu$ and $\boldsymbol{c}^\Sigma$ for every direction, we can:

- Compute the most likely transform as $\boldsymbol{\phi} \boldsymbol{c}^\mu$.
- Visualize the variance map of the coefficients $\mathrm{diag}(\boldsymbol{c}^\Sigma)$, e.g., as a heat map.
- Obtain samples of the field as $\boldsymbol{\phi} A(\boldsymbol{\mu} + \boldsymbol{W}^{-1} \boldsymbol{g})$, where $\boldsymbol{g}$ is a random vector with zero-mean, unit-variance Gaussians at every entry.
- Extract the leading eigenvalues $\{\lambda_i\}$ and eigenvectors $\{\boldsymbol{e}_i\}$ (e.g., with randomized PCA, Rokhlin et al. 2010) to visualize the main modes of variation, e.g., $\boldsymbol{\phi}(\boldsymbol{c}^\mu \pm k\lambda_i \boldsymbol{e}_i)$ with $k \in [-3, 3]$ and $i = 1, 2, 3$.

**Uncertainty-Aware Non-Parametric Transformations.** Best represented by the demons algorithm (Thirion, 1998), most non-parametric registration methods consist of alternating vector field estimation and field regularization (field smoothing). In our case, the field estimation is given by the network prediction, which is fixed, so iterating is not necessary. Instead, we simply convolve the network output with a smoothing kernel $K$ (typically Gaussian) to obtain the transformation. Since this is a linear operation (independently of the choice of $K$), the distribution of the smoothed field remains Gaussian, so we can:

- Compute the most likely transform, which can be efficiently obtained with convolutions: $K \star (\boldsymbol{\sigma}^{-2} \odot \boldsymbol{\mu})/(K \star \boldsymbol{\sigma}^{-2})$, where $\odot$ is the element-wise (Hadamard) product.
- Compute the voxel variance map as: $(K \odot K) \star \boldsymbol{\sigma}^2$.
- Obtain samples of the field as $K \star (\boldsymbol{\mu} + \boldsymbol{W}^{-1} \boldsymbol{g})$, where $\boldsymbol{g}$ is, once again, a random vector with zero-mean, unit-variance Gaussians at every entry.

We note that extracting the leading eigenvalues and eigenvectors in this scenario is also possible, but cannot be effectively done with convolutions and requires using the full expression (Equation (1)).

Throughout the rest of this paper, we refer to the distribution of the fitted transformation (whether it is coefficients $\boldsymbol{c}$ of the smoothed non-parametric field) as the *second level of uncertainty*.

### 3.3 Third level of uncertainty: error bars on downstream tasks

We can further propagate the uncertainty of the transformation to downstream tasks. As an example, we used registration-based segmentation. Given that we can draw samples of the spatial transformation, we can propagate multiple versions of an atlas segmentation (see Figure 1). Each sample from the transformation leads to one possible segmentation map, resulting in different versions of segmentation maps, which can be used to estimate a distribution of labels at every spatial location – and derive its uncertainty using, e.g., the entropy of this distribution. We call this the third level of uncertainty.

### 3.4 Model instantiation

We demonstrate our framework using a simple coordinate-regression DL method, called "Registration by Regression" (RbR), for atlas registration that has been recently proposed (Gopinath et al.,

2024); extension to pairwise registration is straightforward, by regressing displacements from image pairs, rather than atlas coordinates from single images. RbR aims to non-linearly align a given input scan with a target atlas (specifically the MNI template). In this case, $d$ specifies the target coordinates in the MNI space for every voxel in the input scan. Here we describe the losses used to train the baseline RbR model and the additional losses we have used in the training of the extended model.

**Coordinate loss.** For the coordinate loss we use a simple $\ell_2$ loss between the predicted and ground-truth coordinates at every voxel location:

$$L_{coord} = \frac{1}{\sum_n m_n} \sum_{n=1}^{N} m_n \ell_2 [\mathbf{d}_n - F_\theta(\mathbf{I})_n],$$

where $\mathbf{m} = [m_1, ..., m_N]^T$ is a flattened binary foreground mask excluding all non-brain regions and $\mathbf{d}_n$ and $F_\theta(\mathbf{I})_n$ denote the ground-truth and prediction at location $n$ respectively. An $\ell_1$ loss could also be used, but we show in the experiments that this yields worse performance.

**Mask loss.** In addition to regressing the atlas coordinates, the CNN is also trained to predict the brain mask $\mathbf{m}$ as the MNI template we use does not include non-brain structures. For the mask loss we use a combination of a standard cross-entropy and Dice losses with empirically chosen relative weights of 0.25 and 0.75.

$$L_{mask} = \frac{1}{\sum_n m_n} \sum_{n=1}^{N} [0.25 * L_{ce}(m_n, \hat{m}_n) + 0.75 * L_{dice}(m_n, \hat{m}_n)],$$

in which $\hat{\mathbf{m}}$ is the predicted mask and $\mathbf{m}$ is a ground-truth mask, which is also used in the coordinate loss. In this case the cross-entropy and Dice losses are computed over two classes (foreground and background), where the ground-truth and predicted masks are one-hot encoded.

**Atlas segmentation loss (optional).** Given we have a prediction $\hat{\mathbf{d}} = F_\theta(\mathbf{I})$, we can fit a transformation model, as outlined in Section 3.2, to map a segmentation from the target MNI space to the input. This allows us to supervise the network training using a more fine-grained Dice loss at the level of neuroanatomical structures. To this end, we can use the transformation models in Section 3.2 as a differentiable step in the network and train end-to-end using a segmentation loss. Here, the transformation model contains both the linear and non-linear parts. We use a standard Dice loss for the segmentation:

$$L_{seg} = \frac{1}{\sum_n m_n} \sum_{n=1}^{N} L_{dice}(\mathbf{s}_n, \hat{\mathbf{s}}_n),$$

where $\mathbf{s}$ and $\hat{\mathbf{s}}$ are the ground-truth and transformed segmentations respectively, and the Dice loss is now computed over multiple neuroanatomical structures (see Figure 1 for an example segmentation).

**Aleatoric uncertainty loss (optional).** To model the aleatoric uncertainty, we learn to directly predict the means $\hat{\boldsymbol{\mu}}$ and standard deviations $\hat{\boldsymbol{\sigma}}$ of the coordinates. We use the hat to emphasize that these are predictions from the network. When modeling the uncertainty with a Gaussian distribution we aim to minimize the log-likelihood loss w.r.t. $\hat{\boldsymbol{\mu}}$ and $\hat{\boldsymbol{\sigma}}$ given the targets $d$:

$$L_{uncer} = \frac{1}{\sum_n m_n} \sum_{n=1}^{N} \sum_{j=1}^{3} \frac{m_n}{2} \left( \frac{\|d_{n,j} - \hat{\mu}_{n,j}\|^2}{\hat{\sigma}_{n,j}^2} + \log(\hat{\sigma}_{n,j}^2) \right).$$

Here $d_{n,j}$ denotes the ground-truth coordinate for direction $j$ and voxel $n$. In practice, we learn to predict $\log(\hat{\sigma}_{n,j}^2)$ rather than $\hat{\sigma}_{n,j}^2$ directly to map the values to real numbers (not only positive real numbers). Another distribution, such as the Laplace distribution could also be used, and in practice gives similar performance as shown in the experiments.

**Epistemic uncertainty (optional).** To model the epistemic uncertainty, we train the network using Monte Carlo dropout. This amounts to randomly switching off a part of the activation functions in one or more layers. The samples can be generated similarly at test time by making repeated predictions while randomly dropping some of the activation functions. These samples are then summarized as $\hat{\boldsymbol{\mu}}$ and $\hat{\boldsymbol{\sigma}}$ and used in fitting the transformations.

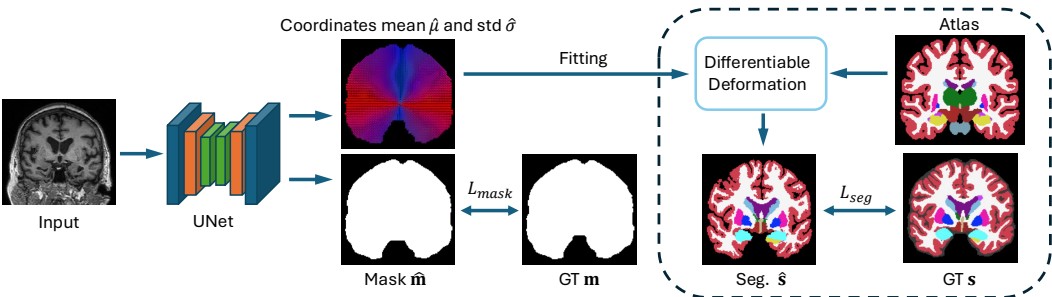

Figure 1: Overview of the training strategy of the proposed method.

**Full loss for different network configurations.** The full loss for the proposed approach is:

$$L_{total} = L_{coord} + \lambda_{mask}L_{mask} + \lambda_{seg}L_{seg} + \lambda_{uncer}L_{uncer}.$$

When training without uncertainty, we simply drop the last term, which models the aleatoric uncertainty. This term is also not included when modeling the epistemic uncertainty, i.e., when training with Monte Carlo dropout.

The overall training framework of the network is illustrated in Figure 1. Given an input scan, the network regresses both the coordinate means $\mu$ and the standard deviations $\sigma$, as well as a foreground mask $\hat{m}$. The optional atlas segmentation loss is denoted by the dashed box.

## 4 EXPERIMENTS

**Training and test data.** We use the same training and test data sets as Gopinath et al. (2024). The training data consists of high-resolution, isotropic, T1-weighted scans of 897 subjects from the HCP dataset (Van Essen et al., 2013) and 1148 subjects from the ADNI (Jack Jr et al., 2008), while the test data set includes the ABIDE (Di Martino et al., 2014) and OASIS3 (LaMontagne et al., 2019) data sets. More details can be found in Appendix A.

**Implementation details.** We use the standard U-net (Ronneberger et al., 2015) as our backbone. More details can be found in Appendix B.

### 4.1 RESULTS

We first present results at every level of uncertainty and qualitatively demonstrate the utility of our approach in a downstream task. We then move on to presenting quantitative results on the registration accuracy using Dice scores as a quality metric. Finally, we show how the different components of the training affect registration accuracy using ablations. Bolded numbers denote significant differences (t-test, $p = 0.05$).

### 4.2 UNCERTAINTY THROUGHOUT THE HIERARCHY

**First-level of uncertainty: epistemic and aleatoric uncertainty.** To ensure that the first-level uncertainties are useful and benefit the registration, they should correlate with the voxel-level coordinate prediction error. The proposed aleatoric uncertainty is simply the re-

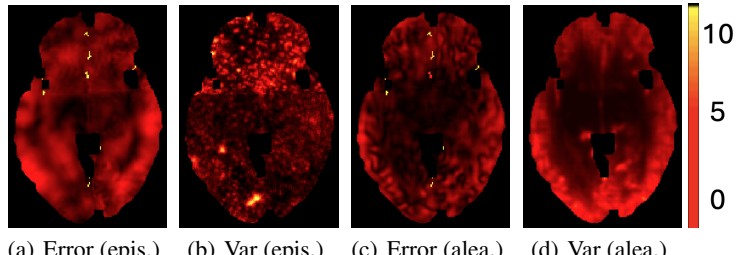

(a) Error (epis.)  (b) Var (epis.)  (c) Error (alea.)  (d) Var (alea.)

Figure 2: Coordinate prediction error in millimeters ($mm$) (Error) and estimated variance in $mm^2$ (Var) for the epistemic (epis.) uncertainty (a, b) and for the aleatoric (alea.) uncertainty (c, d).

gressed $\sigma^2$, whereas for the epistemic uncertainty we train a separate network using dropout layers but without the aleatoric uncertainty loss $L_{uncer}$.

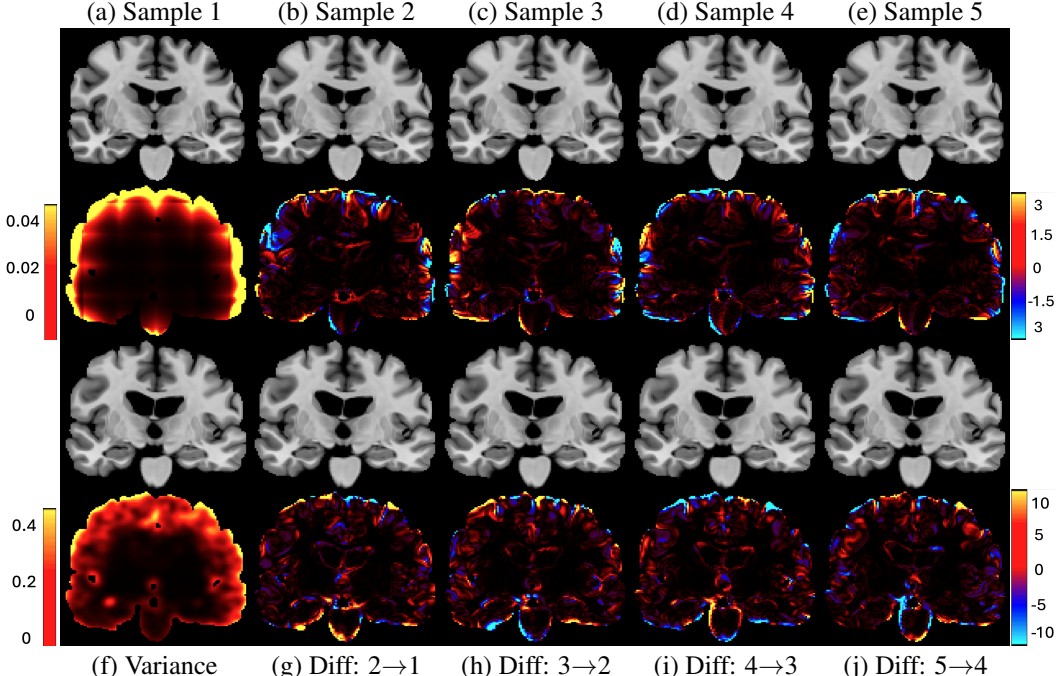

Figure 3: The top row shows samples from the B-spline transformation with their variance and sample-to-sample differences displayed on the second row (from left to right). Similarly, the third row shows samples from the Demons transformation with their variance and sample-to-sample differences shown on the last row (from left to right). Samples and their differences. Diff: 2→1 means the difference between sample 2 and sample 1, and the same applies to others.

In Figure 2, we show, for a single subject, the error between the predicted and ground-truth coordinates along with the voxel-wise variance estimated with epistemic and aleatoric uncertainties. The aleatoric uncertainty highlights the cortex, which is difficult to register, as a region of high variance, whereas the epistemic uncertainty is much more noisy with less structure. We also quantify both the Spearman, which is more robust against outliers, and Pearson correlations between the coordinate error and the variance for both uncertainty approaches over all subjects in the validation set. The correlations for the aleatoric uncertainty are ($\mathbf{0.601 \pm 0.019}$) (Spearman) and ($\mathbf{0.476 \pm 0.021}$) (Pearson), which are significantly higher than the correlations for the epistemic uncertainty ($0.181 \pm 0.017$) (Spearman) and ($0.108 \pm 0.012$) (Pearson). Importantly, the aleatoric uncertainty shows a strong correlation with the coordinate prediction error in absolute terms, which allows effective downweighting of mispredicted coordinates when fitting the transformation, as shown in the next section. Given its superiority, we only consider the aleatoric uncertainty in the subsequent experiments.

**Second-level of uncertainty: uncertainty of transforms.** As outlined in Section 3.2, we can visualize the uncertainty of the parametric and non-parametric transforms by plotting the diagonal of their respective covariance matrices. The heat maps generated in such a way are shown in Figure 4 for the B-spline basis function (10 mm spacing) coefficients and the Demons transformation (3 mm kernel). As for the first level, the largest variances for both transforms coincide with the cortex, which is difficult to register as it is highly folded.

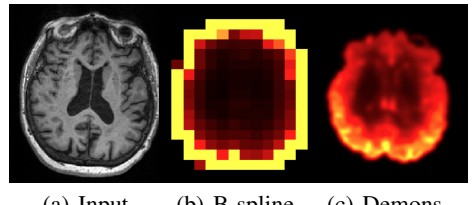

Figure 4: The heat map of uncertainty from B-spline and Demons. Note the B-spline coefficients are upsampled to the image size for visualization.

We note that the B-spline coefficients have very large variances in the background, because it is masked out in the fitting, i.e., there are no target coordinate predictions to match. The Spearman and Pearson correlations between variance and the coordinate prediction error are ($0.619 \pm 0.022$) and ($0.438 \pm 0.020$), respectively, for the Demons transformation. Thus, while the correlations are almost the same as for the network pre-

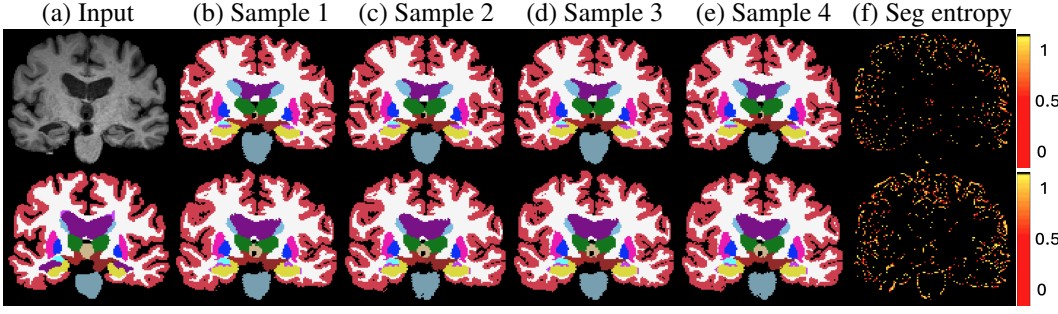

| (a) Input | (b) Sample 1 | (c) Sample 2 | (d) Sample 3 | (e) Sample 4 | (f) Seg entropy |
|---|---|---|---|---|---|

(g) Segmentation  (h) Sample 1  (i) Sample 2  (j) Sample 3  (k) Sample 4  (l) Seg entropy

Figure 5: Example samples from the B-spline transformation (top row, (b-e)) and the Demons transformation (bottom row (h-k)), along with the input scan (a) and the ground-truth segmentation (g). The last column (f, l) show the entropy of the sampled segmentation.

dicted aleatoric uncertainty, the estimate becomes more spatially coherent as shown in Figure 4c. We do not compute the correlation for the variance of the B-spline coefficients as they are defined on a lower resolution grid (10 mm spacing).

**Third-level uncertainty: uncertainty at downstream tasks.** To illustrate how the uncertainty could be used in a downstream task we show samples of atlas deformations and associated propagated segmentations in Figures 3 and 5. Both figures highlight the variability of the samples, which would have a direct effect on any downstream analysis using quantities extracted from the segmentations, e.g., regional volume (Desikan et al., 2009). The sample-to-sample differences, along with the variance, are again concentrated on the cortex. This registration-based uncertainty, when not accounted for, can decrease the power of downstream statistical analyses, or be mistaken for aging effects if the registration errors correlate with age.

## 4.3 REGISTRATION ACCURACY

We quantitatively evaluate the effect of the weighted fitting (second-level of uncertainty) using Dice scores computed between the ground-truth segmentation and the transformed atlas segmentation. Table 1 shows the Dice scores for the affine, B-spline (10 mm), and Demons (3 mm kernel) transformation models fitted with and without uncertainty. The fits without uncertainty are done using only the predicted mean, i.e., effectively setting $\mathbf{W}_j$ to identity matrix. The registration accuracy, measured by Dice, improves for both the affine and B-spline transformations when uncertainty is used, and stays

Table 1: Registration performance for transformations with and without uncertainties.

| Fitting Strategy | ABIDE | OASIS3 |
|---|---|---|
| Affine | $0.718 \pm 0.038$ | $0.673 \pm 0.056$ |
| Affine with uncertainty | $\mathbf{0.730 \pm 0.033}$ | $\mathbf{0.682 \pm 0.055}$ |
| B-Spline 10 mm | $0.782 \pm 0.020$ | $0.750 \pm 0.034$ |
| B-Spline 10 mm with uncertainty | $\mathbf{0.790 \pm 0.019}$ | $\mathbf{0.772 \pm 0.030}$ |
| Demons 3 | $0.799 \pm 0.020$ | $0.783 \pm 0.029$ |
| Demons 3 with uncertainty | $0.799 \pm 0.019$ | $0.783 \pm 0.028$ |

the same for Demons. We further show qualitative examples of segmentations transformed with and without uncertainty for a single subject in Figure 6. The likely reason for the Demons transformation not benefiting from the uncertainty weighting is that we used the Demons transformation in the atlas segmentation loss $L_{seg}$. Thus, the network predicted average coordinates might already be close to optimal and no further weighting is needed. Nevertheless, the qualitative examples show differences for all transformation models, including Demons, when uncertainty weighting is used.

### 4.3.1 ABLATION STUDIES

**Seg. loss and its weight.** In Table 2, we compare the performance of RbR to the proposed approach *without* uncertainty but using the atlas segmentation loss, which was not used in the original RbR model. As expected, the Dice scores of the proposed approach are higher as the model is trained to minimize the Dice loss. We note, however, that incorporating the uncertainty-informed fitting further improves the results as shown above. We further show the effect of changing the loss weight in Table 3.

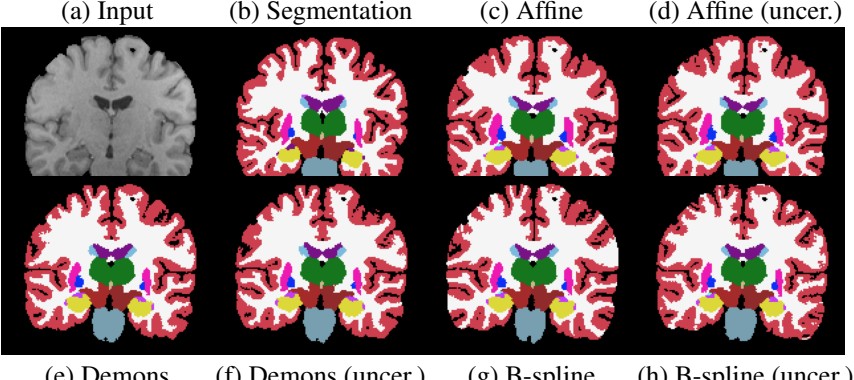

Figure 6: Example transformations fitted with and without uncertainty. (a) and (b) show the input scan and segmentation. (c)-(h) show the transformed segmentations.

Table 2: Comparison between RbR and proposed approach with segmentation loss (average Dice).

| Method | ABIDE | | OASIS3 | |
|---|---|---|---|---|
| | RbR (Gopinath et al., 2024) | Proposed | RbR (Gopinath et al., 2024) | Proposed |
| Affine | $0.712 \pm 0.045$ | $0.715 \pm 0.040$ | $0.667 \pm 0.057$ | $0.669 \pm 0.058$ |
| Demons 1 | $0.734 \pm 0.038$ | $\mathbf{0.764 \pm 0.025}$ | $0.730 \pm 0.026$ | $\mathbf{0.758 \pm 0.025}$ |
| Demons 3 | $0.739 \pm 0.038$ | $\mathbf{0.767 \pm 0.024}$ | $0.728 \pm 0.028$ | $\mathbf{0.759 \pm 0.030}$ |
| Demons 5 | $0.743 \pm 0.039$ | $\mathbf{0.761 \pm 0.026}$ | $0.722 \pm 0.031$ | $\mathbf{0.755 \pm 0.026}$ |
| B-Spline 2.5 | $0.733 \pm 0.038$ | $\mathbf{0.766 \pm 0.024}$ | $0.728 \pm 0.026$ | $\mathbf{0.760 \pm 0.032}$ |
| B-Spline 5 | $0.735 \pm 0.038$ | $\mathbf{0.764 \pm 0.027}$ | $0.725 \pm 0.027$ | $\mathbf{0.757 \pm 0.034}$ |
| B-Spline 10 | $0.738 \pm 0.039$ | $\mathbf{0.758 \pm 0.023}$ | $0.713 \pm 0.032$ | $\mathbf{0.738 \pm 0.029}$ |

**Modeling aleatoric uncertainty.** We model the variance in the uncertainty loss separately for each coordinate direction. Alternatively one can use the same variance for each direction, i.e., same uncertainty for $x, y, z$-coordinates. This is equivalent to an isotropic Gaussian distribution and would simplify the modeling as the uncertainty would be the same for all directions at each location. Table 4 shows the effect of each modeling strategy

Table 3: Ablation study for $\lambda_{seg}$ (average Dice).

| $\lambda_{seg}$ | ABIDE | OASIS3 |
|---|---|---|
| 0.5 | $0.734 \pm 0.053$ | $0.706 \pm 0.043$ |
| 1 | $0.732 \pm 0.064$ | $0.706 \pm 0.053$ |
| 2 | $0.759 \pm 0.035$ | $0.734 \pm 0.036$ |
| 5 | $\mathbf{0.790 \pm 0.019}$ | $\mathbf{0.772 \pm 0.030}$ |
| 10 | $0.771 \pm 0.021$ | $0.769 \pm 0.019$ |

on the Dice score. The results demonstrate that it is important to model the uncertainty in each direction separately. We also ablate the effect of the distribution in Table 5 by comparing the scores when using a Gaussian or a Laplacian. The results are very similar, and in light of this, we chose to use the Gaussian distribution as it has nice theoretical properties as mentioned in Section 3.2.

Table 4: Ablation study for the number of channels to model uncertainty (average Dice).

| # of channels | ABIDE | OASIS3 |
|---|---|---|
| Single channel | $0.714 \pm 0.054$ | $0.687 \pm 0.048$ |
| Three channels | $\mathbf{0.790 \pm 0.019}$ | $\mathbf{0.772 \pm 0.030}$ |

Table 5: Ablation study for distribution used to model uncertainty (average Dice).

| Distribution | ABIDE | OASIS3 |
|---|---|---|
| Gaussian | $0.785 \pm 0.025$ | $0.774 \pm 0.026$ |
| Laplacian | $0.790 \pm 0.019$ | $0.772 \pm 0.030$ |

## 5 CONCLUSION

Uncertainty estimates in DL-based registration approaches are often under-utilized although they provide valuable information about registration accuracy. Here we have proposed a principled approach for propagating the location-level uncertainties (first-level), to fitted transformations (second-level), and finally to downstream analyses (third-level). The results: show that network-estimated *aleatoric* uncertainty correlates well with the coordinate prediction while epistemic uncertainty does not; show that incorporating the aleatoric uncertainty in the transformation fitting improves registration accuracy; and illustrate how the generated samples could benefit downstream analyses. In the future, we aim to extend the framework to pairwise registration of any two input images, and further validate the utility of the uncertainty estimates in aging studies and group comparisons (e.g., between healthy controls and dementia patients).

**Acknowledgments.** We sincerely thank Yaël Balbastre and Bruce Fischl for their helpful discussions. This work is primarily supported by NIH grants 1RF1MH123195, 1R01AG070988, 1R01EB031114, 1UM1MH130981, 1RF1AG080371, 1R21NS138995, and R00 HD10155. Oula Puonti is supported by a grant from Lundbeckfonden (grant number R360–2021–395).

**Ethics Statement.** As the neuroimaging data used in this study is freely available and accessible to anyone after filling out the data usage agreement no ethical approval was needed.

**Reproducibility Statement.** We provide the necessary experimental details in Section 4 as well as Appendix A, Appendix B, including data preparation, training and test data sets, network architecture, and other implementation details. The code to reproduce the results will be made publicly available. Although we cannot redistribute the data, all data sets are freely available for download after filling out the data usage agreement.

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

# A TRAINING AND TEST DATA

We use the same training and test data sets as Gopinath et al. (2024). The training data consists of high-resolution, isotropic, T1-weighted scans of 897 subjects from the HCP dataset (Van Essen et al., 2013) and 1148 subjects from the ADNI (Jack Jr et al., 2008). These two data sets provide a good mix of older (ADNI) and younger (HCP) subjects capturing a large range of spatial deformation. The training scans were resampled to 1mm$^3$ isotropic resolution, masked, and registered to the ICBM 2009b Nonlinear Symmetric MNI template using NiftyReg (Modat et al., 2010). Specifically, we first ran the block matching algorithm for affine registration (`reg_aladin`) with the `-noSym` option, and subsequently used the fast free-form deformation algorithm (`f3d`) to compute the nonlinear registration. `f3d` was run in diffeomorphic mode (`-vel`) and local normalized correlation coefficient ($\sigma = 5$) as similarity metric. The processing time for the whole dataset was less than 24h on a 64-core desktop. To obtain the ground-truth anatomical segmentations, we used SynthSeg Billot et al. (2023) which segments the input scan to 32 different structures. The SynthSeg segmentation is also used to create the brain mask. The total training set consists of the coordinates, segmentations, and brain masks, and is split 80/20% between training and validation.

The test data set consists of two public data sets, ABIDE (Di Martino et al., 2014) and OASIS3 (LaMontagne et al., 2019), both consisting of high-resolution, isotropic, T1 scans. Similar to the training set, we have both younger (ABIDE) and older (OASIS3) subjects for testing. We selected the first 100 scans from both data sets for evaluation so that the test data set matches that of Gopinath et al. (2024). The test data is processed exactly the same way as the training data sets, yielding the coordinates, segmentations and brain masks. The ground truth registration from NiftyReg was visually quality controlled on both data sets to check for obvious registration errors on both data sets. None of the registrations needed to be excluded.

# B IMPLEMENTATION DETAILS

We use the standard U-net (Ronneberger et al., 2015) as our backbone. It has four resolution levels with two convolutional layers (comprising $3 \times 3 \times 3$ convolutions and a ReLu) followed by $2 \times 2 \times 2$ max pooling (in the encoder) or upconvolution (decoder). The final activation layer is linear, to regress the atlas coordinates in decimeters (which roughly normalizes them from -1 to 1). We empirically set $\lambda_{mask} = 0.5$, $\lambda_{seg} = 5$ and $\lambda_{uncer} = 0.1$. The learning rate is 0.01. The parameters are chosen via the validation performances. The input MRI scan(s) $\mathbf{I}$ undergo intensity augmentation with blurring, bias fields, and noise. Furthermore, we also use spatial augmentation, both affine and nonlinear, which is applied to MRI scans as well as the segmentations $\mathbf{s}$ and masks $\mathbf{m}$ to ensure spatial correspondence.

# C MORE ABLATION STUDIES

**Choice of coordinate loss.** For the coordinate regression, we can either adopt a $L1$ or a $L2$ loss as a distance measure. $L1$ is often used because of its higher robustness, but $L2$ is usually faster to converge when training. The Dice score difference between the losses is shown in Table 6. The $L2$ loss performs slightly better than the $L1$ loss.

Table 6: Ablation study for regression loss (average Dice).

| Regression loss | ABIDE | OASIS3 |
|---|---|---|
| $L1$ | $0.783 \pm 0.024$ | $0.763 \pm 0.016$ |
| $L2$ | $\mathbf{0.790 \pm 0.019}$ | $\mathbf{0.772 \pm 0.030}$ |

Table 7: Ablation study for fitting strategy (average Dice).

| Deformation strategy | ABIDE | OASIS3 |
|---|---|---|
| B-spline | $0.710 \pm 0.022$ | $0.700 \pm 0.024$ |
| Demons | $\mathbf{0.790 \pm 0.019}$ | $\mathbf{0.772 \pm 0.030}$ |

**Choice of transformation during training.** For the atlas segmentation loss, we need to transform the atlas segmentation to input space. We compare B-splines (10 mm spacing) to Demons (3 mm kernel) in Table 7. The non-parametric transformation performs better, however we did not test all possible control point spacings for B-splines or smoothing kernels for Demons. It is possible that another parameter combination would yield even better performance.

## D WHY DOES THE UNCERTAINTY IN THE FIT TRANSFORM NOT IMPACT THE RESULT FROM THE DEMONS TRANSFORMATION MODEL?

In the original submission, we speculated that there was little to gain by fitting with uncertainty when using Demons, because the model training used a Demons fit already in the atlas segmentation loss (see Section 4.3), which limits the margin for improvement when you are fitting this same model at test time.

To further validate our assumption, we conducted two additional experiments:

1. Use a B-spline transformation during training instead of the Demons. If our speculation about the limited improvement at test time is correct, fitting the B-spline transformation with uncertainty at test time should show only a marginal improvement compared to fitting without uncertainty. As shown in Table 8 this is indeed the case: the Dice scores for the test-time fits with and without uncertainty are almost exactly the same.

Table 8: Registration performance for transformations with and without uncertainty if training with B-spline.

| Fitting Strategy | ABIDE | OASIS3 |
|---|---|---|
| w/o uncertainty | $0.710 \pm 0.025$ | $0.699 \pm 0.023$ |
| w/ uncertainty | $0.710 \pm 0.022$ | $0.700 \pm 0.024$ |

2. Alternatively, we trained the model using a direct deformation, i.e., interpolating using the predicted coordinates directly without using a transformation model, in the atlas segmentation loss. Now, as shown in Table 9, the Demons transformation does benefit from fitting with using the uncertainty estimates, which further supports our initial suspicion that the improvement for the Demons model was marginal because it was used during training.

Table 9: Registration performance for transformations with and without uncertainties if trained with direct deformation.

| Fitting Strategy | ABIDE | OASIS3 |
|---|---|---|
| w/o uncertainty | $0.767 \pm 0.024$ | $0.759 \pm 0.030$ |
| w/ uncertainty | $\mathbf{0.799 \pm 0.019}$ | $\mathbf{0.783 \pm 0.028}$ |

Based on these results, it seems that the uncertainty estimates are helpful in generalizing the fitting accuracy across transformation models (beyond the one used during training).

## E COMPUTATIONAL COST

In our experiments, training took approximately 1.2h per epoch on an NVIDIA RTX A6000 GPU, with convergence typically requiring 50 epochs. During inference, the average runtime for a single sample ($176 \times 256 \times 256$) was approximately 15 minutes to generate all the results, which includes coordinate prediction and, fitting all of the transformation models (affine, B-spline, Demons) with and without uncertainty. During the inference, most of the time is spent on the uncertainty-aware Bspline fitting, in which we compute the inverse of the high dimension matrix (please refer to Section 3.2). We note that the fitting and sampling steps can be further optimized for speed, and that the inference for a single model, e.g., Demons, can be done considerably faster.

## F LIMITATIONS

Here we have only included two representative non-linear transformation models and restricted the application to registration to a fixed atlas space. Due to computational limitations, we did not compute the closed-form solution for the B-spline coefficients for spacings $< 10$mm. Using Dice scores as the only proxy for comparing registration accuracy gives only a partial understanding of the quality of the registrations (Rohlfing, 2012). Finally, the usefulness of the third-level uncertainties is presented only qualitatively, and not quantifying it in a relevant application is left for future work.

