# OpenReview forum: "Hierarchical Uncertainty Estimation for Learning-based Registration in Neuroimaging"
_ICLR.cc/2025/Conference — ICLR 2025 Poster_

### Official Review · Reviewer_Eicy · 2024-10-30

**Soundness:** 3
**Presentation:** 3
**Contribution:** 2
**Rating:** 6
**Confidence:** 3

**Summary:**

In this paper, the authors propose a registration framework that includes uncertainty estimation at three different levels (network output, transform models' parameters and downstream tasks). They show that uncertainty estimates correlate with registration errors and that estimating uncertainty can improve registration performance. The approach is applied to brain MRI registration.

**Strengths:**

- The framework proposed is flexible and can be applied to a broad range of registration techniques.
- The authors performed diverse experiments to validate their approach.
- The results are mostly convincing.
- The paper is well written and related works well introduced.

**Weaknesses:**

- The ground truths considered originate from automatic registration and segmentation approaches but it is not mentioned whether quality control was performed and so to which extent they can be relied on.
- The experiments regarding the downstream task are limited to qualitative results and the fact that the segmentations are not overlaid on the T1w image only allows assessing differences between the segmentation maps but not whether one matches better the anatomy than another one.

**Questions:**

- In Figure 2, what does the error correspond to exactly?
- Could the authors comment on the run time cost of their approach or other considerations regarding its practical application?

===
- The authors already mention many related works but they could also consider:
    - Chen J et al. "A survey on deep learning in medical image registration: New technologies, uncertainty, evaluation metrics, and beyond." arXiv preprint arXiv:2307.15615 (2023).
    - Zhang X et al. "Heteroscedastic Uncertainty Estimation Framework for Unsupervised Registration." International Conference on Medical Image Computing and Computer-Assisted Intervention. Cham: Springer Nature Switzerland, 2024.
And comment especially on the second one.

---

> ### Author Response · Authors · 2024-11-24
> **Response to the comments of Reviewer Eicy**
>
> We thank the reviewer for the thorough review and comments. Below we address specific concerns one-by-one.
>
> **Q1**. Is quality control performed on the ground truths?
>
> **A1**: Yes, we have visually checked all the registrations as explained in Reviewer X1vX Q9. We have now also clarified this in Appendix A (lines 830-832 in the revised version).
>
> **Q2**. The experiments regarding the downstream task are limited to qualitative results; Overlaying the segmentation with T1w images to show which segmentation is better instead of only showing the difference.
>
> **A2**: Figure 6 shows examples of warped segmentations using the different transformation models (affine, B-spline, Demons) fitted with and without uncertainty estimates, and Table 1 shows the corresponding quantitative results. In this case, the most accurate segmentation (based on Table 1) is from the Demons model (panel e and f) followed by the B-spline fit with uncertainty (panel h). Additionally, we show the ground truth segmentation in panel b, which can be used as a reference when comparing the warped segmentations from the transformation models.
>
> Figures 4 & 5 show individual samples drawn from the B-spline and Demons transformation models, along with a reference segmentation in Figure 5 (panel g). Here, we decided to visualize the differences between the samples to highlight areas of high uncertainty. We note that for downstream analyses, we would not use any of the individual samples, but rather compute estimates, e.g., the volume variation of a structure, using all of them. In this case, if one sample is locally more accurate than another is less relevant for the estimates. We plan to quantify the practical benefit of the downstream uncertainties in future work.
>
> **Q3**. In Figure 2, what does the error correspond to exactly?
>
> **A3**: In Figure 2, we show, for a single subject, the error between the predicted and ground-truth coordinates in millimeters along with the voxel-wise variance ($mm^2$) estimated with epistemic and aleatoric uncertainties.
>
> **Q4**. Could the authors comment on the run time cost of their approach or other considerations regarding its practical application?
>
> **A4**: In our experiments, training took approximately 1.2h per epoch on an NVIDIA RTX A6000 GPU, with convergence typically requiring 50 epochs. During inference, the average runtime for a single sample ($176 \times 256 \times 256$) was approximately 15 minutes to generate all the results, which includes coordinate prediction and, fitting all of the transformation models (affine, B-spline, Demons) with and without uncertainty. During the inference, most of the time is spent on the uncertainty-aware Bspline fitting, in which we compute the inverse of the high dimension matrix (please refer to Section 3.2). We note that the fitting and sampling steps can be further optimized for speed, and that the inference for a single model, e.g., Demons, can be done considerably faster. We have included the computational costs in the appendix (Appendix D in the revised version).
>
> **Q5**. The authors already mention many related works but they could also consider:
> 1. Chen J et al. "A survey on deep learning in medical image registration: New technologies, uncertainty, evaluation metrics, and beyond." arXiv preprint arXiv:2307.15615 (2023).
>
> 2. Zhang X et al. "Heteroscedastic Uncertainty Estimation Framework for Unsupervised Registration." International Conference on Medical Image Computing and Computer-Assisted Intervention. Cham: Springer Nature Switzerland, 2024.
>
> **A5**: We thank the reviewer for the suggestion. The revised version includes a reference to the survey paper (which some readers may indeed find useful) and also discusses the connection between our work and Zhang et al. (lines 154-161 in the revised version) Their work is parallel to our first level of uncertainty (i.e., we could replace our aleatoric model with theirs); our contribution lies in the hierarchical model of uncertainty we build on top.

---

> > ### Comment · Reviewer_Eicy · 2024-11-27
> >
> > I'd like to thank the authors for their answers and clarifications. I will leave my score untouched as it was already positive and the comments I had were minor.

---

> > > ### Author Response · Authors · 2024-11-27
> > > **Thank you for your support**
> > >
> > > Dear Reviewer Eicy,
> > >
> > > Thank you very much for your response and valuable comments to improve our manuscript!
> > >
> > > Best,
> > >
> > > Authors of paper #7885

---

### Official Review · Reviewer_X1vX · 2024-11-03

**Soundness:** 2
**Presentation:** 2
**Contribution:** 2
**Rating:** 5
**Confidence:** 4

**Summary:**

This paper introduces a framework to integrate uncertainty estimation into deep learning-based image registration for neuroimaging. By propagating epistemic and aleatoric uncertainties from voxel-level predictions to transformation models and downstream tasks, the framework enhances registration accuracy and reliability. Experiments show that this uncertainty-aware approach can boost brain MRI registration performance.

**Strengths:**

1. This paper addresses an important problem in neuroimaging, aiming to use uncertainty estimation to enhance the accuracy and reliability of deep learning-based registration.
2. The figures and explanations are clear and well-organized, which makes complex ideas easier to understand.
3. The empirical findings are insightful. The authors compare their proposed method with other different methods.

**Weaknesses:**

1. The presentation could be improved. The background, motivation, and knowledge gap are not clearly explained, making it challenging to follow the paper’s purpose and direction. Although the study is about why uncertainty estimation matters in brain registration, it doesn’t make a strong case for why this is important. There isn’t enough support or reasoning behind this focus, which leaves readers unsure about the value or impact of the work.
2. The experiments are insufficient. This paper only compares its method to a single approach, RbR. Although RbR was released in 2024, it is still unpublished and available only on arXiv, which makes it less robust as a baseline for comparison. Including more established and published baselines would be essential to provide a solid foundation for evaluating the effectiveness of the proposed approach.
3. From the experimental results (e.g., Table 1), the introduction of uncertainty estimation does not appear to provide a statistically significant improvement to the model. This makes it difficult to assess its effectiveness and raises questions about the practical value of incorporating uncertainty estimation into the approach.
4. Most of the latest work in brain registration focuses on unsupervised learning, largely due to the high cost of collecting accurate transformation ground-truth for high-dimensional images (e.g., 3D MRI). However, this paper still stay on a supervised learning approach, making it seem less aligned with current trends and potentially less practical for real-world applications where labeled data is limited.
5. The evaluation criteria lack validity. This paper collects ground-truth transformations using NiftyReg, a tool introduced around 15 years ago. Since then, more state-of-the-art methods have been shown to outperform NiftyReg, indicating that the ground-truth it provides may be inaccurate. This undermines the reliability of the results presented in the paper and raises concerns about the credibility of its findings.
6. The novelty of this work is limited. The main contribution appears to be the addition of an uncertainty loss term, while the other loss terms and network architecture are based on existing work. This incremental improvement does not significantly advance the field, as much of the framework relies on previously established methods.

**Questions:**

1. Why is uncertainty estimation important in neuroimaging registration? What specific benefits does it bring, and why is it necessary?
2. For modeling epistemic uncertainty, why did the author choose to use Monte Carlo dropout, and how exactly was it implemented?
3. Given that the deformation field (non-linear transformation) is a high-dimensional tensor, how do authors verify the accuracy of the ground-truth labels when collecting them? As I understand, NiftyReg is a registration method—if this method is used to generate ground-truth labels, does that imply the proposed method can only perform as well as NiftyReg at best?
4. The total training loss includes multiple terms marked as optional in the paper. Are these terms truly necessary? Was an ablation study conducted to assess their impact on performance?

---

> ### Author Response · Authors · 2024-11-24
> **Response to the comments of Reviewer X1vX (1/2)**
>
> We sincerely thank you very much for your detailed comments. We really appreciate your efforts to help improve the quality of this manuscript. We will improve the presentation according to the suggestions. Below we address other specific concerns one-by-one.
>
> **Q1**. Why is uncertainty estimation so important in brain registration?
>
> **A1**: Uncertainty estimation is crucial in brain registration because it enhances both the **reliability** and **interpretability** of the registration process, which is foundational for many neuroimaging analyses. Here are some key reasons why it is so important:
>
> 1. Error Localization
>
> Brain structures vary widely in size, shape, and texture across individuals. Registration uncertainty can pinpoint regions where errors are likely, such as in aligning brain regions that are small (e.g., the hypothalamus) or have poorly defined boundaries (e.g., the nucleus accumbens), thus enabling more targeted validation or adjustment.
>
> 2. Inter-Subject and Population Variability
>
> In group studies, individual anatomical variability may complicate registration. Uncertainty estimation captures areas where variability exceeds the model's ability to confidently align images, facilitating better handling of population-level differences (e.g., downweighting subjects that are poorly registered in a certain area).
>
> 3. Algorithmic Transparency
>
> Modern registration methods, particularly those using machine learning, can behave as "opaque boxes." Uncertainty estimation adds transparency by highlighting where the model is less confident, helping researchers trust the algorithm's outputs.
>
> 4. Risk Mitigation in Medical Applications
>
> In high-stakes scenarios, such as radiotherapy targeting or brain surgery navigation, understanding registration uncertainty helps mitigate risks by avoiding over-reliance on potentially inaccurate image alignment.
>
> 5. Clinical Decision Support
>
> Uncertainty estimation provides a quantifiable measure of confidence in the alignment of brain images. In clinical settings, such as surgical planning or diagnosis, understanding where registration results are less reliable helps guide clinicians in making more informed decisions.
>
> 6. Impact on Downstream Analysis
>
> Many neuroimaging studies rely on registered images for tasks like statistical analysis, segmentation, or functional mapping. Errors in registration propagate to these analyses. Uncertainty estimation allows researchers to weigh or exclude data from unreliable regions, improving the robustness of downstream results.
>
> In summary, uncertainty estimation not only strengthens the credibility of brain registration results but also enhances their utility in both research and clinical applications, making it a cornerstone of robust and interpretable neuroimaging workflows. We have expanded the uncertainty estimation paragraph in the introduction (lines 64-72 in the revised version), with some of the most crucial points listed above.
>
> **Q2**. The experiments are insufficient. It only compares with RbR (which is unpublished).
>
> **A2**: Our contribution is not a new registration method, but rather equipping existing methods with a hierarchical model of uncertainty. This is why our experiments explore a broad array of model fits, rather than competing registration algorithms.
> In terms of the base method: we chose RbR (which is now published https://link.springer.com/chapter/10.1007/978-3-031-73480-9_16) because, being a direct coordinate regression method, it makes it straightforward to illustrate the application of our proposed framework.
>
> **Q3**. The introduction of uncertainty estimation does not appear to provide a statistically significant improvement to the model (Table 1).
>
> **A3**: Significant improvements (marked in **bold** in the table) were obtained for the affine and B-spline models, on both tested datasets. We have now added further experiments to explain why the Demons model did not show a significant improvement, please see Q1 of Reviewer ofnx above.
>
> We would also like to emphasize that the improvement due to the uncertainty weighting is only one of the benefits of our framework: the uncertainty estimates are also useful *per se* (for the reasons described in Q1), and also yield uncertainty at other levels including downstream tasks.
>
> **Q4**. A supervised learning setting makes it seem less aligned with current trends and potentially less practical for real-world applications where labeled data is limited.
>
> **A4**: The supervision occurs at the coordinate level, and NiftyReg is used to obtain the target coordinates automatically (i.e., “for free”). Therefore, our framework is highly practical as it does not require human labeling to produce huge amounts of training data. The regression framework also has the advantage of greatly facilitating the modeling of aleatoric uncertainty (see “Aleatoric uncertainty loss”).

---

> ### Author Response · Authors · 2024-11-24
> **Response to the comments of Reviewer X1vX (2/2)**
>
> **Q5**. Why use NiftyReg, a tool introduced around 15 years ago.
>
> **A5**: NiftyReg has improved over the years, and includes a combination of principled deformation models and regularizers that produce excellent results. For example, the authors of the state-of-the-art learning method “SynthMorph” (Hoffmann et al., Imaging Neuroscience, 2024) showed that NiftyReg is as accurate as their method, if not more, for 1mm isotropic brain MRI scans (see Figures 5 and 8 in the paper).
>
> **Q6**. The novelty of this work is limited. The main contribution appears to be the addition of an uncertainty loss term, while the other loss terms and network architecture are based on existing work.
>
> **A6**: As explained in Q1 and (to a lesser extent) Q3, the novelty of our work lies in the hierarchical framework to model registration uncertainty, rather than specific architectures or losses.
>
> **Q7**. For modeling epistemic uncertainty, why did the author choose to use Monte Carlo dropout, and how exactly was it implemented?
>
> **A7**: MC Dropout is simple, effective, and widespread. We add dropout layers to the end of the first and second blocks of our UNet.
>
> **Q8**. Given that the deformation field (non-linear transformation) is a high-dimensional tensor, how do authors verify the accuracy of the ground-truth labels when collecting them?
>
> **A8**: The ground truth registrations were visually quality controlled (which has been clarified in Appendix A lines 830-832 in the revised version); none of them had obvious large errors. Either way, we emphasize that these ground truth coordinates were used only for training, not for evaluation (which relies on Dice scores). Therefore, the quality of this ground truth is irrelevant towards the validity of our results.
>
> **Q9**. The total training loss includes multiple terms marked as optional in the paper. Are these terms truly necessary? Was an ablation study conducted to assess their impact on performance?
>
> **A9**: Our submission features over a page of results on ablation studies, which we believe is a considerable amount, given the page limit. Specifically, Table 3 highlights the impact of segmentation loss, while Tables 4 & 5 illustrate how incorporating uncertainty loss influences registration performance. For additional details, please also refer to the other ablation study tables provided.

---

> ### Comment · Reviewer_X1vX · 2024-11-26
>
> Thank you for your response and the clarification. I have updated my score accordingly.

---

> ### Author Response · Authors · 2024-11-26
> **Thank you for your response**
>
> Dear Reviewer X1vX,
>
> Thank you very much for your response! Your valuable comments have significantly enhanced the quality of our work. We kindly ask if you have any additional questions or concerns regarding our manuscript. We would be happy to discuss further and look forward to addressing any further feedback you might have.
>
> Best,
>
> Authors of paper #7885

---

> > ### Author Response · Authors · 2024-12-01
> > **Follow-up**
> >
> > Dear Reviewer X1vX,
> >
> > We hope this message finds you well. We kindly wanted to check if you had any additional comments or suggestions to improve our manuscript.
> >
> > Thank you for your time and feedback!
> >
> > Best,
> >
> > Authors of paper #7885

---

### Official Review · Reviewer_ofnx · 2024-11-03

**Soundness:** 4
**Presentation:** 4
**Contribution:** 3
**Rating:** 8
**Confidence:** 4

**Summary:**

Authors present a method to estimate the uncertainty in medical image registration at 3 different stages: (1) on the estimate of the distribution of the deformation field at each voxel (assuming this is Gaussian and gathering a mean and standard deviation per point), (2) on the distribution of the fitted transformation, and (3) on the distribution of possible outcomes on the downstream task. The uncertainty from the first level is used in the transformation fitting step weighting down contributions from less certain pixels. By drawing samples of the fitted transformation, a distribution of results for the same downstream task is generated which is used to estimate the 3rd level of uncertainty. Authors demonstrate their approach in the context of registration-based Brain image segmentation where the given input image is deformed to match the standard MNI atlas and the labels from the atlas are propagated to the deformed image.

**Strengths:**

* Authors clearly lay out their proposed framework describing the 3 level of uncertainty they they aim to model.
* The results are presented such that each level of uncertainty is evaluated - this makes it easy for the reader to better understand the Authors important contribution.
* Authors provide analysis of the estimates of uncertainty, and demonstrate that the aleotoric uncertainty corresponds better with coordinate prediction error at the first level.
* The figures in the paper are very useful for visualizing the variations that arise from sampling the fitted transform - they demonstrate the fundamental problem with biomedical image registration in that the results from a downstream task are highly dependend on the transformation paramaters, but the uncertainty can be successfully quantified.

**Weaknesses:**

* The addition of uncertainty in the fit transform did not impact the result from the Demons transformation model. Authors explain that this is likely because the loss from the Demons method is used in the segmentation loss and that leads to the average coordinates landing close to the optimum. This is unclear - please can Authors expand on this and also elaborate on whether they would expect to see improvements if they used an alternative to Demons?

**Questions:**

* The panel labels in Figure 2 are badly formatted - please make these clearer.

---

> ### Author Response · Authors · 2024-11-24
> **Response to the comments of Reviewer ofnx**
>
> Thanks for your positive feedback! We hope to address your concerns with additional experiments and the following discussions.
>
> **Q1**. Why does the uncertainty in the fit transform not impact the result from the Demons transformation model?
>
> **A1**: In the original submission, we speculated that there was little to gain by fitting with uncertainty when using Demons, because the model training used a Demons fit already in the atlas segmentation loss (see Section 4.3), which limits the margin for improvement when you are fitting this same model at test time.
>
> To further validate our assumption, we conducted two additional experiments:
> 1. Use a B-spline transformation during training instead of the Demons. If our speculation about the limited improvement at test time is correct, fitting the B-spline transformation with uncertainty at test time should show only a marginal improvement compared to fitting without uncertainty. As shown in the table below this is indeed the case: the Dice scores for the test-time fits with and without uncertainty are almost exactly the same.
>
> | Fitting Strategy | ABIDE | OASIS3 |
> |:------:|:------:|:------:|
> |  w/o uncertainty   |   0.710 $\pm$ 0.025 |    0.699 $\pm$ 0.023   |
> |  w/ uncertainty  |  0.710 $\pm$ 0.022  |   0.700 $\pm$ 0.024   |
>
> 2. Alternatively, we trained the model using a direct deformation, i.e., interpolating using the predicted coordinates directly without using a transformation model, in the atlas segmentation loss. Now, as shown in the table below, the Demons transformation does benefit from fitting with using the uncertainty estimates, which further supports our initial suspicion that the improvement for the Demons model was marginal because it was used during training.
>
> | Fitting Strategy | ABIDE | OASIS3 |
> |:------:|:------:|:------:|
> |  w/o uncertainty   |   0.767 $\pm$ 0.024 |    0.759 $\pm$ 0.030   |
> |  w/ uncertainty  |  **0.799 $\pm$ 0.019**  |   **0.783 $\pm$ 0.028**   |
>
> Based on these results, it seems that the uncertainty estimates are helpful in generalizing the fitting accuracy across transformation models (beyond the one used during training). We have included these new results in the appendix (Appendix C in the revised version).
>
> **Q2**. The panel labels in Figure 2 are badly formatted - please make these clearer.
>
> **A2**: Thanks for pointing this out. We have reformatted Figure 2 (lines 367-370 in the revised version).

---

> > ### Comment · Reviewer_ofnx · 2024-11-25
> > **Acknowledgement of Author Response**
> >
> > Thank you to the Authors for their considered response. This Reviewer appreciates the additional experiments that aim to validate the hypothesis resulting in minimal gains using the Demons fit at atlas segmentation and test time. It is encouraging to see that using the uncertainty estimates are helpful in generalizing fitting accuracy beyond the models using during training.
> >
> > Furthermore, the response addressing the concerns of Reviewer `X1vX` is thorough, and highlights the novel contributions of the paper in terms of the hierarchical framework for effectively incorporating uncertainty into the process.
> >
> > This Reviewer's score remains unchanged, and I recommend acceptance of the paper.

---

> > > ### Author Response · Authors · 2024-11-25
> > > **Thanks for your support**
> > >
> > > Dear Reviewer ofnx,
> > >
> > > Thank you very much for your positive feedback and valuable comments to improve our manuscript!
> > >
> > > Best,
> > >
> > > Authors of paper #7885

---

### Author Response · Authors · 2024-11-24
**General response**

We thank the reviewers for their time and insightful feedback. We have revised the paper to clarify the main motivation of our approach (see Reviewer X1vX Q1, lines 64-72 in the revised version) and added new experiments, which further demonstrate the benefit of including uncertainty for fitting the transformations (see Reviewer ofnx Q1, Reviewer X1vX Q3, and the revised appendix). We would like to emphasize that our aim is not to propose a new registration method, but rather to develop a general uncertainty propagation framework that can be combined with different deep learning-based registration approaches.

We have updated the revised version to reflect the modifications (highlighted in blue). Below we address specific concerns one-by-one.

---

### Meta-Review · Area_Chair_JW1R · 2024-12-20

**Metareview:**

This work developed a method to estimate the uncertainty in medical image registration at 3 different stages: (1) on the estimate of the distribution of the deformation field at each voxel, (2) on the distribution of the fitted transformation, and (3) on the distribution of possible outcomes on the downstream task. The uncertainty from the first level is used in the transformation fitting step weighting down contributions from less certain pixels. By drawing samples of the fitted transformation, a distribution of results for the same downstream task is generated which is used to estimate the 3rd level of uncertainty.


This work has three reviewers. Two reviewers are positive and the other reivew is negative to accept this work. After the rebuttal discussion between the reviewers and the authors, two positive reviewers appreciates the additional experiments and clarifications, and keep their positive rating. The negative also keep the final rating of 5, but reduce its confidence on reviewing this work.

**Additional Comments On Reviewer Discussion:**

After checking the responses from the authors, two positive reviewers still agree to accept this work. And the negative also upgraded the score accordingly.  The final ratings of three reviewers are 8, 6, and 5. The two positive reviewers think that the answers, the experiments, and the clarifications at the rebuttal well address their concerns and are happy to accept this work. For the negative review, he also find many pros from the author response. Although the negative reviewer still has some concerns and keeps the negative rating, but he or she reduce the reviewing confidence. After checking the paper and the rebuttal contents, I agree with these two positive reviewers to accept it due to the novelity of this method, and many pros of this work.

---

### Decision · Program_Chairs · 2025-01-22

Accept (Poster)